# Assessing the EU Climate and Energy Policy Priorities for Transport and Mobility through the Analysis of User-Generated Social Media Content Based on Text-Mining Techniques

Anastasia Nikolaidou [1], Aristomenis Kopsacheilis [1], Nikolaos Gavanas [2,*] and Ioannis Politis [1]

1   Transport Engineering Laboratory Aristotle, Department of Civil Engineering, Faculty of Engineering, University of Thessaloniki, 54124 Thessaloniki, Greece; nikolaid@civil.auth.gr (A.N.); kopsacheilis@civil.auth.gr (A.K.); pol@civil.auth.gr (I.P.)
2   Department of Planning and Regional Development, School of Engineering, University of Thessaly, 38334 Volos, Greece
*   Correspondence: ngavanas@uth.gr

**Abstract:** For over three decades, the European Union's (EU) transport policy has aimed at fostering environmental sustainability and energy efficiency. Since 2015, European policymakers have focused more on three key sustainable development goals: decarbonizing the transport system, promoting low-emission mobility solutions, and transitioning to renewable and alternative fuels. To effectively communicate priorities and engage stakeholders, EU policymakers regularly use social media platforms like Twitter (now known as X). This active discourse involves policymakers, industrial stakeholders, the media, and the public, offering insights into the role of transport policy in addressing climate change and energy transition challenges. The current research endeavors to track and analyze the evolution of user-generated content related to climate change, energy transition, and smart mobility on Twitter from 2011 to 2021. This research uses text-mining and social network analysis techniques to quantitatively and qualitatively assess the dynamics of relevant EU policies and their effects. The study's findings can be used to establish a robust monitoring and evaluation framework at the EU and national levels. This framework will assess the effectiveness of communicating strategic priorities for sustainable transport development. It also holds potential for application in other sectors, broadening its impact.

**Keywords:** social media; climate change; energy transition; transport; mobility; data mining; topic modeling; social network analysis

## 1. Introduction

The 21st century has seen the rise of social media, which has transformed communication worldwide. These platforms, accessible across devices, enable global interactions. Examples include Twitter (now known as X; the authors of this research decided to use the former platform name because the data used were generated before the renaming), Facebook, Instagram, and LinkedIn [1,2]. Users connect, share interests, and express opinions, contributing to diverse content. Social media's versatility caters to various purposes, from marketing to political campaigns [3,4].

Social media revolutionizes information dissemination, offering unparalleled accessibility. It influences political views through abundant ideas and opinions [5]. Globally, it attracts attention for effecting societal change and promoting causes like energy conservation and political engagement. Its accessibility empowers individuals and organizations to raise awareness and mobilize support globally [6].

The academic literature predominantly focuses on using social media platforms in political communication, particularly within or by European institutions, national European Union (EU) representatives, and political figures at various levels [7]. Research typically

examines two main aspects: content analysis, which studies post properties and topics, and citizen participation, which is assessed through sentiment analysis and the frequency of reactions [8]. This approach comprehensively explains social media's role in political discourse and engagement [9].

In the context of European institutions, social media presents an opportunity to engage with European citizens directly, bypassing traditional media intermediaries such as journalists or editors who filter information [9]. This development has generated optimism among EU organizations, believing that social media channels will foster closer connections between public organizations and the public [10–13]. Specifically, social media is a direct communication tool for EU transport policy, aiming to accelerate the shift towards sustainable mobility and address ongoing challenges within the transport sector [6]. Despite the undoubted progress over the last decades, the EU transport policy still faces great challenges regarding climate change and energy transition. Transport accounts for approximately 25% of Europe's greenhouse gas (GHG) emissions [14]. Road transport alone represents approximately 70% of the total GHG emissions from transport. Moreover, the EU transport sector depends on oil to cover about 90% of its energy needs, with road transport accounting for 47% of oil demand [15].

However, a little over a decade after public institutions took their first steps on social media, research points towards a relatively unsuccessful use of such technologies in Western democracies [12,16–18]. A fundamental problem identified in this line of research is what could be called the "presence-attention gap" on social media: While many public institutions are present on social media, they receive relatively little attention on these platforms [5,6].

Social media represents an underutilized resource for driving policy action and societal change [19]. However, the vast array of platforms and analytical techniques offers numerous avenues for providing valuable insights into policymaking and decision-making processes. For example, social media content offers a real-time gauge of the impact of policy interventions, providing policymakers with immediate feedback on their effectiveness. Moreover, network analysis of social media platforms allows for tracking influence dynamics, revealing how individuals can shape and be shaped by their social connections. Such analysis delves into various facets of social relationships, including identifying key influencers, characterizing specific communities, and mapping the flow of information [20]. In essence, harnessing the power of social media and its analytical capabilities presents a wealth of opportunities for enhancing policy formulation and decision-making processes.

In this context, the purpose of the current research is to analyze the relation of the EU policies for transport and mobility with the issues of climate change and energy transition, as well as to investigate the way that these policies are communicated to the stakeholders through social media platforms. More specifically, the research is based on data mining from the microblogging platform Twitter from 2011 to 2021. The following research questions are framed:

- How did the discussion between different types of Twitter users (policymakers, industrial stakeholders, the media, and the general public) regarding the role of transport policy in tackling the challenges of climate change and energy transition evolve during the period under consideration?
- What are the emerging topics being identified in the context of the broader climate change debate? How do these specific topics evolve over time? Furthermore, how is the discussion around these topics organized by different categories of users?
- How do new ideas spread through a network, and who are the influencers that start the spread?

The remainder of this paper is structured as follows. The next section delves into how public organizations utilize social media, specifically in the EU. It also highlights the use of text-mining and social network analysis techniques and tools to better comprehend social media analysis about policy decision-making, communication, public engagement, and related areas. Following Section 2, the methodology for the text mining and analysis

conducted in the current research is presented. Section 4 refers to presenting the results from processing collected information and data. That section is followed by a discussion of results and, finally, a presentation of conclusive remarks and plans for follow-up research.

## 2. Literature Review

The ongoing digital revolution has significantly reshaped the governance landscape, particularly evident in the profound changes brought about by social media platforms. These platforms have become essential tools for public administrations seeking to modernize their processes and foster closer ties with citizens. In Europe, where concerns about communication deficits between the EU and the public have long been voiced, leveraging social media has emerged as a strategic imperative [13,21–27].

Public sector entities within the European Union have increasingly turned to platforms like Twitter, Facebook, and Instagram to bridge the gap with citizens, aiming to amplify their reach and foster meaningful engagement. This shift towards digital communication channels reflects a broader organizational transformation, altering internal processes and redefining the dynamics between governments and the public. Furthermore, it extends to the very nature of civil service roles, influencing how officials conduct their duties [28–31].

The European Union agencies have a crucial role in executing EU policies and enhancing the EU's image and reputation. For several agencies, external communication is a paramount activity. Through communication, they can provide information and research-based evidence on particular issues, trends, and challenges faced during the EU policy-making process and implementation. Their information is meant to inform policymakers and the public [6]. European public institutions leverage social media platforms to disseminate information, engage with citizens, and promote EU policies in a more accessible and interactive manner. Examples include (a) European Commission's Twitter updates and direct dialogue with citizens, (b) European Parliament's Facebook Live sessions with Members of the European Parliament (MEPs) to discuss policy issues, (c) European External Action Service's Instagram Stories to show behind the scenes of EU diplomacy, (d) European Union Agency for Fundamental Rights' targeted social media campaigns to raise awareness about fundamental rights issues, and (e) European Central Bank's Twitter chats with experts to increase understanding of the bank's role in the EU.

Amid this digital evolution, one recurrent challenge stands out: ensuring that pertinent information does not get lost in the noise of the online sphere but instead effectively reaches the intended audience. This challenge is particularly poignant for EU institutions, which have been historically criticized for perceived communication deficiencies. Recognizing the pivotal role of agencies in addressing these shortcomings, the EU emphasizes the significance of external communication in conveying policy-related information to policymakers and the public [7,8].

In academia and practice alike, there is a burgeoning interest in understanding the efficacy and impact of public administrations' use of social media. Scholars and practitioners advocate for a departure from traditional communication paradigms, urging European institutions to embrace social media's benefits in connecting with citizens. However, while much attention has been given to the political communication, there remains a gap in understanding how social media contributes to the broader social construction of European policy, integration, and identity [32–35].

Unraveling the complexities of social media attention unveils a dual focus: longstanding attention, indicative of an organization's enduring audience base, and temporal attention, reflecting the immediate engagement with individual messages. Both facets are integral for public agencies, enabling them to maintain a steady dialogue with their audience while also disseminating important updates swiftly across wider networks [18,36–38].

This discourse on social media attention prompts further inquiry into the factors influencing audience engagement and institutional communication strategies. By exploring these dimensions, we can gain deeper insights into how social media dynamics shape public discourse, policy formation, and the construction of a European public sphere.

Text mining is crucial in unlocking valuable insights from the vast text data generated on social media platforms, enabling organizations to make informed decisions, understand user behavior, and engage effectively with their audience. Text mining, also known as text analytics or natural language processing (NLP), involves extracting meaningful insights and patterns from unstructured text data. Social media platforms generate vast amounts of text data in posts, comments, tweets, and messages, making text-mining techniques invaluable for analyzing and understanding social media content [39–41].

Applying text-mining techniques can help create a robust theoretical framework that connects social media analysis with policy decision-making and public engagement theories, enhancing the study's context and providing deeper insights into the dynamics of social media's influence on policymaking processes and public engagement. Text-mining techniques can be used to conceptualize how social media platforms function as public spheres, according to the Sphere Model of Public Discourse proposed by Jürgen Habermas, where citizens engage in discussions, deliberation, and opinion formation. Additionally, based on policy cycle and multi-flow models, social media analysis can identify emerging policy issues, inform policy formulation through public input, facilitate policy implementation by engaging stakeholders, and contribute to policy evaluation through feedback and monitoring. Drawing from public engagement theories, such as Arnstein's ladder of citizen participation or the concept of deliberative democracy, studies can assess the quality and extent of public engagement facilitated by social media platforms. By examining the level of citizen involvement, dialogue, and influence in social media discussions, studies can evaluate the effectiveness of public engagement efforts in shaping policy outcomes [42–45].

Text mining is applied in social media through different techniques, such as sentiment analysis, topic modeling, named entity recognition (NER), user profiling and behavior analysis, anomaly and event detection, and social network analysis (SNA) [46–51].

In our current research, we are mainly engaged in topic modeling and social network analysis of social media data. Topic modeling algorithms can be applied to identify underlying topics or themes in social media conversations. This helps uncover trending topics, detect emerging issues, and segment discussions based on common themes. Overall, topic modeling enables public organizations to extract actionable insights from diverse unstructured text data sources, supporting evidence-based decision-making, stakeholder engagement, and effective governance. By leveraging topic modeling techniques, public organizations can enhance their capacity to understand complex issues, respond to citizen needs, and achieve their mission of serving the public interest [52–55]. Social network analysis involves analyzing user relationships and interactions in social media networks. Text mining can aid in extracting network structures, identifying influential users or communities, and studying information diffusion patterns within social networks. Overall, social network analysis enables public organizations to understand the complex social structures and dynamics that shape their interactions with stakeholders, facilitate collaboration, and influence decision-making processes. By leveraging SNA techniques, organizations can enhance their capacity for networked governance, stakeholder engagement, and collaborative problem-solving to pursue public interest objectives [56–59].

Despite social media's growing prominence, there needs to be more research analyzing European organizations' social media accounts to gauge their outreach levels. Researchers often grapple with quantifying influence on Twitter, a leading micro-blogging platform used by European organizations, utilizing metrics such as followers, retweets, and mentions. However, studies reveal complexities in interpreting these metrics; for instance, the number of followers alone does not necessarily denote influence, as demonstrated by patterns of reciprocity and homophily. Moreover, while European agencies increasingly embrace social media for communication and engagement, there needs to be more systematic understanding regarding their platform preferences, purposes, and performance measurements. Current social media metrics often lack depth and fail to justify resource investments adequately [60–63].

Effective social media programs must align with agency missions and objectives, necessitating institutionalized monitoring and measurement. Existing analytics tools like Hootsuite or Google Analytics may only partially cater to the unique goals of European agencies [64].

Although social media has enabled direct and transparent communication between agencies and citizens, there is a pressing need for well-designed performance metrics to assess their effectiveness. Despite social media's growing importance, there remains to be a gap in understanding when and how different European agencies utilize social media platforms and the corresponding outcomes. Therefore, this study aims to investigate the extent and manner of social media usage by European agencies related to climate change, energy transition, and smart mobility, particularly on Twitter, and propose practical performance metrics for future evaluation.

## 3. Materials and Methods

This section presents the crucial elements of the methodology implemented in the present research. Figure 1 illustrates the basic steps of the methodology, from reviewing EU transport policy milestone documents and collecting social media data, to extracting predominant topics in datasets and analyzing the spread of influence on social networks regarding EU climate and energy policy priorities for transport and mobility.

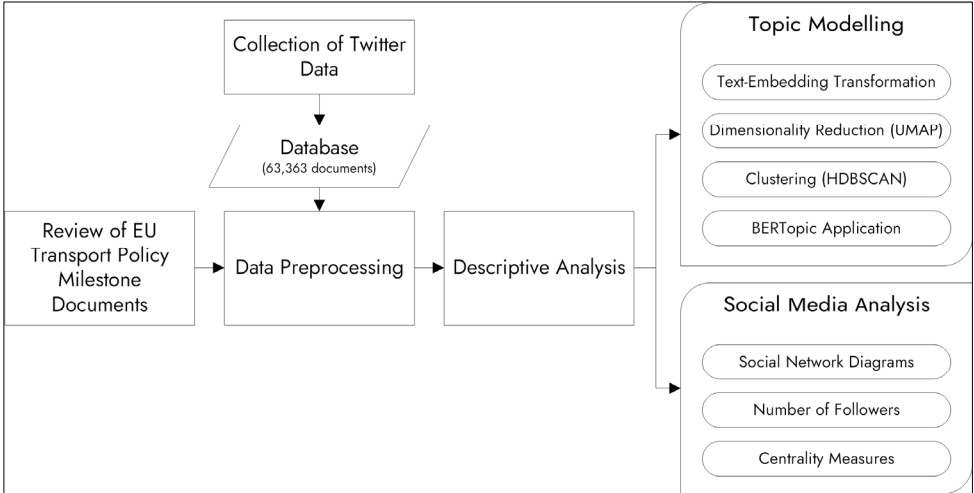

**Figure 1.** Methodology flowchart.

Section 3.1 of the present research reviews milestone documents of EU transport policy. Section 3.2 and Section 3.3 describe how Twitter data were collected and processed for this project. Section 3.4 presents natural language processing techniques used in topic modeling. Section 3.5 discusses the concept of social network analysis used to examine the social connections of Twitter users discussing specific topics.

### 3.1. Preliminary Analysis: Review of EU Transport Policy Milestone Documents

With the purpose of setting the background for the understanding and analysis of the concepts and findings from the current research, a comprehensive review of EU transport policy milestone documents was conducted, aiming to point out the main policy recommendations. As milestone documents are considered the high-level strategic frameworks periodically communicated by the European Commission in the last three decades, formulating the official common transport policy, the review includes the overarching transport policy strategies with a focus on the strategies for sustainable (urban) mobility. The urban level was selected to be part of the review due to the significant role of cities in decarbonization. It should be noted here that approximately 70% of GHG emissions globally are produced in cities [65].

*3.2. Data Collection*

The study aims to track and analyze the evolution of user-generated content on Twitter related to climate change, energy transition, and smart mobility.

The microblogging platform Twitter was chosen for its suitability for the following reasons. First, according to recent studies, Twitter is one of the leading social networks worldwide, with many active users [66]. Secondly, EU policymakers use Twitter extensively as a communication tool to promote EU initiatives and priorities [67]. Thirdly, Twitter data are open and accessible to everyone through the Twitter Application Programming Interface (API). Unlike other social media sites like Facebook, Twitter data can be gathered using the Academic Research product track to collect historical tweets [68]. Lastly, Twitter data contain information related to posts (tweet text, time stamp, number of re-tweets, etc.) and users (author ID, number of followers and following, etc.), enabling a more comprehensive analysis of the sample. Despite recent ownership changes, Twitter remains one of the most accessible platforms for data collection, although access to historical data for research purposes is now somewhat limited [69].

The research covers the period from 1 January 2011, to 31 December 2021, to ensure consistency with the timeline of EU policy documents, which are presented in Section 3.1 of the paper. During the data collection stage, keywords related to the research subject were used. The appropriate keywords were selected based on monitoring specific terms presented in users' posts that link transport policies with sustainable mobility, climate change, and energy transition priorities. Four (4) different data categories were created by organizing groups of keywords related to transport, climate change, and energy transition. The keywords for each category correspond to the keywords referring to the inter-relation between transport, climate change, and energy transition, which were identified by the review of main documents that define the EU policy for transport and mobility. These documents, which are also included as bibliographical references to the current paper, are the European Commission's main strategic document for the current Programming Period, i.e., the European Green Deal published in 2019 [70], as well as its reference documents, i.e., the United Nations' 2030 Agenda for Sustainable Development [71] and the Paris Agreement of 2015 [72]; the main transport policy frameworks of the last two decades, i.e., the Transport White Papers of 2001 [73] and 2011 [74], and the Sustainable and Smart Mobility Strategy of 2020 [14]; and the main urban mobility policy frameworks of the same period, i.e., the Urban Mobility Green Paper of 2007 [75], the strategy for competitive and resource-efficient urban mobility of 2013, and the New EU Urban Mobility Framework of 2021 [76]. The first category includes general terms of transport and mobility related to sustainable development, while the second contains terms associated with the specific challenge of climate change. The third category includes terms that describe policy priorities to address climate change. The fourth and final category describes the relationship between climate change and transport policy priorities, explicitly focusing on electromobility, connectivity, automation, and shared mobility [77]. Table 1 illustrates the categories formulated and the selected keywords for each category.

Twitter offers different API endpoints to access various types of data. One of these endpoints is the "Search Tweets" (GET/2/tweets/search/recent) that can be used to crawl text data. This endpoint allows users to search for recent tweets based on specific criteria. To collect data from Twitter, we used the Python programming language, version 3.9, and specific code libraries such as Tweepy and search-tweets python. Furthermore, we used the Academic Research product track to collect historical tweets. Once we collected tweets from the API, we processed the data as needed, including filtering and cleaning. We then stored the data in a suitable format such as CSV, JSON, and database [68].

**Table 1.** Data collection keywords and categories.

| Category | Keyword 1 | Keyword 2 | Description |
|---|---|---|---|
| 1 | mobility; transport; smart mobility; smart transport | sustainable development; sustainability | Transport and sustainable development in general |
| 2 | mobility; transport | climate change; climate urgency; global warming; carbon footprint | Transport and climate change |
| 3 | mobility; transport | decarbonization; climate neutral; zero carbon; low carbon; energy transition; clean energy; green energy; climate mitigation; climate adaptation; climate resilience; climate resilient | Transport and policy priorities for tackling climate change |
| 4 | decarbonization; climate neutral; zero carbon; low carbon; energy transition; clean energy; green energy; climate mitigation; climate adaptation; climate resilience; climate resilient | autonomous transport; autonomous mobility; automated transport; automated mobility; transport automation; connected transport; connected mobility; transport connectivity; electrification; electromobility; electric vehicle; hydrogen; fuel cell; mobility as a service; shared mobility; shared transport | Policy priorities for tackling climate change and transport policy priorities |

### 3.3. Data Preprocessing

To ensure the accuracy and relevance of our data, we implemented various measures to eliminate redundancy from our dataset, using Python version 3.9 for all necessary tasks.

Initially, we filtered out non-English tweets from our dataset, removing 34,812 such tweets. Subsequently, we conducted a search for duplicate tweets, resulting in the exclusion of an additional 44,043 tweets. Given our research focus on EU mobility policies, we only retained tweets posted by users within the EU. This entailed manual processing to remove tweets originating from countries outside the EU. Notably, as our analysis covered the period preceding Brexit (2011–2021), tweets from the United Kingdom were also included. Following these steps, our final dataset comprised 63,373 tweets.

Further data preprocessing involved the removal of unwanted elements from each tweet. This encompassed filtering out web page links, retweets, numbers, hashtags, punctuation marks, special characters, and white spaces to obtain a more concise dataset. For this purpose, we utilized the Natural Language Toolkit (NLTK) in Python to eliminate stop words like "to", "by", "so", and "it", and similar terms that do not significantly contribute to the tweet's meaning. Additionally, two-character words that remained in certain tweets were also removed.

### 3.4. BERTopic

In order to explore the association of EU's climate change, energy transition, and decarbonization strategies with the corresponding policies regarding transport, we need to explore the semantics of the collected documents in our database. Since such a process is very demanding, especially when it comes to thousands of text excerpts, like in our case, a manual analysis and interpretation of each excerpt is not optimal. Instead, we decided to follow a more global approach in order to explore the semantics of the collected tweets and determine the underlying topics.

For our analysis, we used the BERTopic library through the Python programming environment. Although topic modeling can be implemented using several popular techniques, such as Latent Dirichlet Allocation (LDA) or Non-Negative Matrix Factorization (NMF), we opted to use BERTopic due to its overall superior performance in extracting insights from Twitter data [78]. Another advantage of BERTopic is its modular character, which gives analysts the ability to customize it by combining different models/techniques for the various steps of the analysis (text embedding, dimensionality reduction, clustering,

etc.). Ultimately, these techniques can be inserted into the BERTopic model as parameters, thus ensuring the optimal performance of the model. The foundation of BERTopic lies in the BERT (Bidirectional Encoder Representations from Transformers) natural language model. The advantage of BERT against other natural language models refers to its ability to bidirectionally process text data in a document, thus identifying context more accurately. The pre-training of BERT on a large number of documents and the subsequent availability of embeddings for words in different contexts provides a basis for the development of strong topic models, for any use case.

The first step in the process of applying BERTopic in a given corpus of text data refers to the transformation of text into embeddings, i.e., numerical representations. In order to generate embeddings that reflect the meaning of each as accurately as possible, a preprocessing step of the documents needs to be performed beforehand. More specifically, during this stage, the segmentation of each document into smaller "tokens" is performed to assist the embedding model in detecting terms in each document. In our case, we exploited the "CountVectorizer" function to perform the preprocessing of our documents. Although other topic models require further preprocessing steps, such as the removal of stop words or lemmatization, these are not prerequisites for the BERTopic model, since stop words and derivatives of words assist the model in determining the actual context of each sentence. In our analysis, we exploited pre-trained transformer models for creating dense vectors of each document, and, more specifically, the "all-MiniLM-L6-v2" embedding model. This choice was made due to the model's low computational requirements and due to the accurate depiction of the semantic meaning of each document, as was proven in the topic clustering later in the methodology. However, due to the utilization of high dimensional vectors from the transformer models utilized for the representation of the meaning of each document (384 dimensions in our case), a visual inspection of the results of the topic analysis is not possible. To counter this issue, dimensionality reduction algorithms were applied to the embeddings in order to project their information in two (2) or three (3) dimensions without losing any contextual details. The modular character of BERTopic allows for the application of several dimensionality reduction algorithms, such as t-distributed stochastic neighbor embedding (t-SNE) and uniform manifold approximation and projection (UMAP). For our analysis, we choose to use the UMAP model, which according to the literature has the ability to distinguish groups of data more clearly [79], while being significantly faster in computation time compared to t-SNE [80]. By following a trial-and-error approach and by inspecting the output of the model, we selected the optimal value for the number of neighbors (equal to 100), thus ensuring that the model can represent the global context of the corpus as well as keep local characteristics.

The determination of the contextual similarity of documents and the subsequent formation of potential clusters is the next step in the methodology. The modularity of BERTopic again provides the opportunity of cluster model selection among several alternatives. In our case, we considered two (2) of the most popular clustering models, k-means and HDBSCAN. The combination of the latter model (HDBSCAN) with BERTopic has been characterized by superior performance in comparison to the k-means model and was thus selected in our methodology. By assessing condensed tree plots for several values of the parameters of the model, we managed to determine the optimal combination of values (minimum cluster size equal to 500 and minimum number of samples per cluster equal to 50), which led to the best clustering formation in terms of cluster number, cluster complexity, and cluster overlapping.

After the selection and formulation of the appropriate parameters and models regarding the embedding creation, dimensionality reduction, and clustering of our documents, we inserted these models as parameters in the BERTopic model. Subsequently, the model was trained on the embeddings created and, by taking the pre-identified clusters from the HDBSCAN model as guidance, the documents of the dataset were categorized in the final topics. The generated topics were then inspected for cohesion regarding the allocation of documents and semantic similarity between topics. In cases where the misallocation of

documents in a topic occurred or similar topics were detected, the process was repeated by modifying the parameters of the embedding or the clustering process, respectively, until the optimal number of topics was generated.

One of the advantages of BERTopic compared to other topic analysis techniques is the fact that in cases where the algorithm does not determine a definite association of a document to a topic, it flags it as an outlier, thus giving a more robust view of the underlying topics in a corpus. Despite that, the detected outliers can still be allocated to an identified topic through the application of one (1) of the strategies or methods available through BERTopic.

*3.5. Social Media Influence Analysis*

Twitter users create a network when they interact with each other by mentioning or replying. This network can be analyzed using graphs to gain valuable insights about its size, structure, and nodes. Social network analysis (SNA) is a field of study that examines social structures through networks and graph theory. It explores relationships between individuals, groups, or organizations, representing them as nodes (or vertices) and their connections as edges (or links).

Centrality measures and community detection techniques are powerful tools in social network analysis. However, their effectiveness is amplified when combined with network visualization tools. These tools enable the visualization of social networks, making it easy to explore and analyze network structures. Nodes are typically represented as points, while edges are represented as lines connecting the nodes. This visual representation can reveal patterns, clusters, and essential nodes within the network, making the analysis process more intuitive and efficient.

Social network analysis is a valuable tool with many applications across various fields, including sociology, anthropology, psychology, epidemiology, organizational studies, and computer science. Researchers rely on this method to better understand social phenomena, pinpoint noteworthy individuals or groups, analyze information flow, and forecast behaviors within social networks. When applied to Twitter, this method can unveil users' connectivity and highlight influential users and their connections by measuring their centrality [20,81,82].

This paper focuses on categorizing tweets based on specific keywords. Each category creates a separate network comprising Twitter users and their tweets. The network is represented by nodes that signify users and edges that represent the tweets and connect the nodes. To determine the importance of the users in the network for disseminating a message, it was considered appropriate to calculate a number of measures in addition to the representation of the network. The following measures were calculated for each data collection category in the current study:

- Number of followers: A Twitter account's number of followers is a crucial factor in determining its reach and influence. Accounts with more followers tend to have a wider audience and, therefore, a more significant impact. The total number of followers can be used to identify the most popular users on Twitter [83].
- Eigenvector centrality: In many cases, having a connection to a popular person is more valuable than having a connection to someone not well-connected. The eigenvector centrality network metric considers the number of connections a vertex has (i.e., its degree) and the importance of the vertices it is connected to. Essentially, it considers "how many people you know" and "who you know" [20]. For a given graph $G := (V, E)$ with $|V|$ vertices let $A = a_{v,t})$ be the adjacency matrix, i.e., $a_{v,t} = 1$ if vertex $v$ is linked to vertex $t$, and $a_{v,t} = 0$ otherwise. The relative centrality score, $x_v$, of vertex $v$ can be defined as [84]:

$$x_v = \frac{1}{\lambda} \sum_{t \in M(v)} x_t = \frac{1}{\lambda} \sum_{t \in V} a_{v,t} x_t \qquad (1)$$

where *M(v)* is the set of neighbors of *v* and $\lambda$ is a constant. With a small rearrangement, this can be rewritten in vector notation as the eigenvector equation:

$$Ax = \lambda x \tag{2}$$

- Betweenness centrality: Betweenness centrality measures how much a vertex acts as a bridge in a network. When there are no relationships between social actors or group actors in a network, those in structural holes obtain strategic benefits such as control, access to new information, and resource brokerage. Actors who fill these structural holes are considered desirable relationship partners because of their advantageous structural position [20]. The betweenness of a vertex *v* in a graph *G: = (V, E)* with *V* vertices is computed as follows [85]:

$$Betweeness\ (v) = \sum_{s \neq v \neq t \epsilon V} \frac{\sigma_{st}\ (v)}{\sigma_{st}} \tag{3}$$

  where $\sigma_{st}$ is the total number of shortest paths from node *s* to node *t* and $\sigma_{st}\ (v)$ is the number of those paths that pass through *v*.
- Based on the above, the present research identified three (3) measures to determine Twitter influencers. The first measure is based on the number of overall Twitter followers and identifies global influencers. The second and third measures help to identify local influencers within specific networks by analyzing who is frequently mentioned and retweeted.

Social network analysis was performed using the NodeXL tool, Pro Academic/Non-Profit version, an open-source software platform for analyzing and visualizing social networks [20].

## 4. Results

### 4.1. Comprehensive Review of EU Transport Policy Milestone Documents

The EU's first Transport White Paper [86] was published three decades ago and can be linked to the overall focus on sustainable development policies in Europe, such as the introduction of the legal basis for a common European environmental policy in the same period [87], and the leading role of the European transport sector in the global debate for the promotion of sustainable mobility. The white paper highlights that transport growth generates environmental problems and "green-house effects" and that the problem can no longer be addressed by isolated interventions but through a global approach. It outlines the priority for cleaner vehicles, fuels, and engines, and for the promotion of intramodality. Since then, the efforts of the European transport policy to promote sustainable mobility have been constantly increasing. The Transport White Paper of 2001 depicts the challenges of congestion and oil dependency in the transport sector in an era of enlargement for the EU [73]. The specific policy framework promotes technological innovation towards the development of alternative fuels, such as biofuels, natural gas, and hydrogen, as well as hybrid and battery electric engines, and foresees the potential of shared mobility solutions. Focusing on urban mobility, the Transport Green Paper [75] relates the intense environmental challenges of cities with the need for more sustainable travel choices and driving behavior, in combination with the improvement in engines and fuels. The next Transport White Paper, which was published in 2011, presents a roadmap to achieve 10 goals for increasing competitiveness, reducing fuel dependency, improving energy efficiency, and achieving a 60% reduction in GHG emissions from transport by 2050 [74]. In this context, the policy framework discusses that the deployment of clean vehicles can be accelerated through appropriate infrastructure, such as refueling/recharging stations, and by capitalizing on the potential of cities to become the test beds for electric, hydrogen, and hybrid technologies. Smart mobility services are also promoted by the white paper to enhance transport's socio-economic and environmental sustainability. It is worth mentioning that the European Commission published in 2013, as part of the Urban Mobility

Package, a strategy for urban mobility which annexes the concept of Sustainable Urban Mobility Plans (SUMPs) [88].

In the current programming period, i.e., 2021–2027, the European Green Deal constitutes the EU's strategic framework for sustainable development in all sectors, including transport [70]. The Green Deal presents a roadmap for the implementation of the United Nation's (UN) Sustainable Development Goals (SDGs) [71] and the Paris Agreement [72]. Based on the priority of the Green Deal for "accelerating the shift towards sustainable and smart mobility", the Sustainable and Smart Mobility Strategy sets the goal of a 90% reduction in GHG emissions from transport by 2050 compared to 1990 [14]. The strategy adopts the Green Deal's approach for presenting specific targets by 2030 and 2050, including carbon-neutral scheduled collective travel of under 500 km, wide-scale deployment of automated mobility, and at least 30 million zero-emission vehicles by 2030 (while nearly all road vehicles should be zero emission by 2050). The combination of policies and measures is prioritized, such as the take-up of battery and hydrogen fuel-cell electric and other types of zero-emission vehicles, the development of refueling/charging infrastructure with integration into the smart energy grid, and the promotion of shared and active mobility as part of an integrated and smart multimodal urban transport system. Furthermore, the New EU Urban Mobility Framework outlines eight fields of priority for the transition to safe, smart, resilient, and zero-emission urban mobility for all by promoting active and collective transport solutions [76]. The framework also refers to the potential of clean energy and electrification, but also connected and automated mobility, and new mobility services to support the decarbonization of urban transport systems.

Figure 2 illustrates a timeline of EU transport policy documents analyzed in this study.

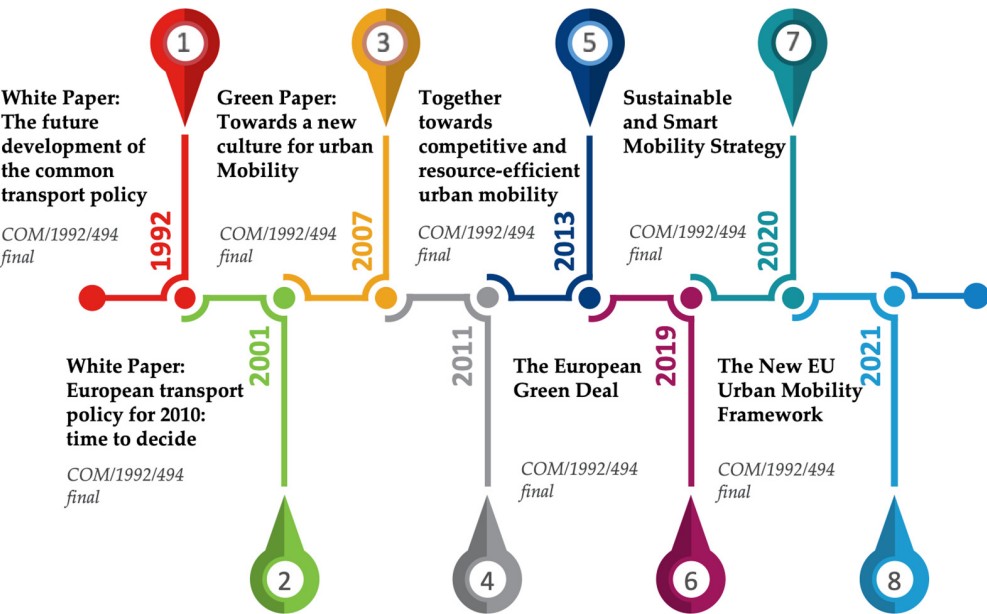

**Figure 2.** Primary policy documents adopted by the European Commission (1992–2021).

### 4.2. Descriptive Statistics of Twitter Data Sample

During data collection, we gathered 231,390 Twitter posts for our research. After applying specific criteria outlined in the data preprocessing stage, a final sample of 63,373 tweets remained for the period 2011–2021. Regarding unique users, our analysis identified 28,614 individuals who posted on Twitter about the research topic in question.

Over 50% of the sample consists of posts that other users have retweeted. The average number of retweets for the entire sample is 2.63, meaning that, on average, other network users share a post on the given subject more than twice. This rate indicates that these posts have relatively low popularity, as the number of times a Twitter text is retweeted indicates what is trending. A post that is retweeted multiple times within a short period goes viral.

Based on the analysis of different types of posts on Twitter, it was found that most of the messages posted by users include one or more references to other users. These references are called mentions, and they make up approximately 40% of all posts. Simple user posts, known as tweets, make up over 33% of the posts. The remaining 27% of posts are messages that are replies to other users, known as "reply to". Mentions and replies to a post can indicate user attention and capture community engagement and outreach.

Figure 3 shows the total number of tweets collected per year and the percentage of tweets belonging to each data content category. The thickness of each line in the chart represents the quantity of data collected each year, while the different colors represent the categories based on which the data was collected.

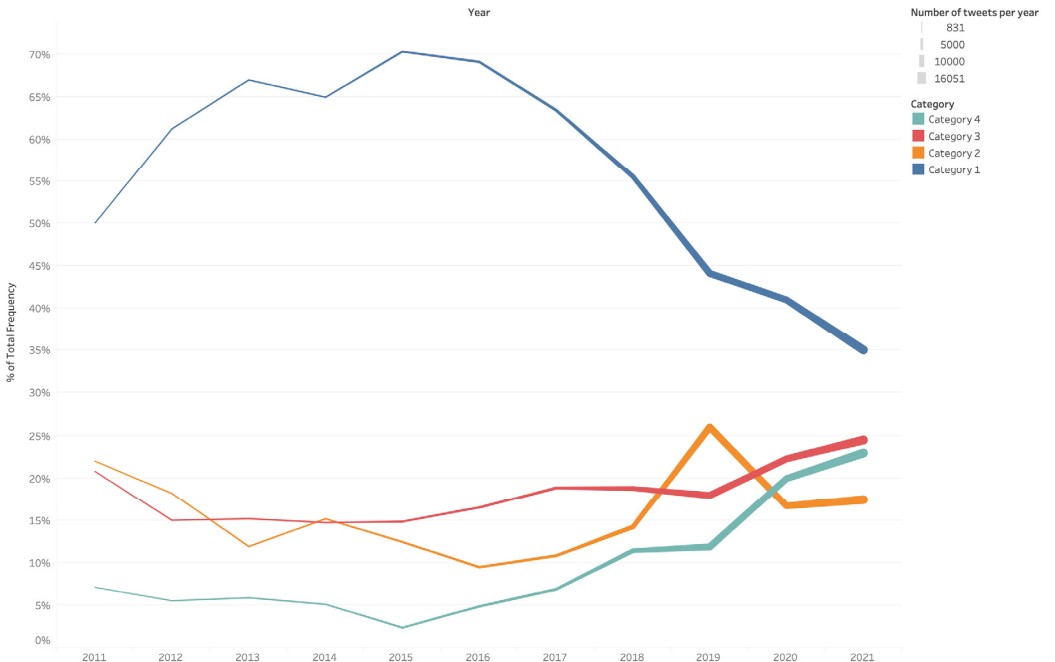

**Figure 3.** Total number of "tweets" per year and per category.

Based on the findings depicted in Figure 3, a consistent increase in the total number of collected tweets is evident over time. This upward trend, observed annually, can be attributed to the growing popularity of social media platforms. However, it is likely to be also influenced by heightened interest in climate change. Particularly notable spikes occurred in 2012, 2015, 2018, and 2019, coinciding with significant EU policy milestones such as the implementation of the 2011 Transport White Paper, the adoption of the United Nations Sustainable Development Goals and the Paris Agreement, and the initiation of the European Green Deal during the current EU Programming Period. In 2020, there was a slight decrease of approximately 5% in user engagement regarding this subject, likely due to the EU's heightened focus on addressing the COVID-19 pandemic, which shifted users' attention and posting behavior.

During the data collection phase of the methodology, publications were categorized into four (4) content categories designed to explore various facets of transport and sustainable development, as presented in Table 1. Category 1 exhibited the highest percentage of publications, primarily centered on general transport and sustainable development themes. Conversely, Category 4 had the lowest number of user posts overall, focusing mainly on transport policy priorities aimed at addressing climate change.

From 2011 to 2015, there was a steady increase in posts within Category 1 compared to the other categories. However, since 2015, the number of Category 1 posts has notably decreased, while there has been an increase in those of the other categories, notably Category 4. This shift reflects the EU's commitment to achieving specific climate change targets

by 2030 and 2050, as Category 4 concentrates on transport and mobility policies geared towards decarbonization and energy transition goals. Consequently, we have witnessed a transition from emphasizing the relationship between transport and sustainable development/climate change (Categories 1 and 2), to understanding the role of transport in achieving decarbonization and energy transition targets and promoting specific transport and mobility policies to meet these targets (Categories 3 and 4).

Based on the analysis, the average number of retweets per post is the highest in Category 2, while the lowest average number of retweets per post is found in Categories 1 and 4. This implies that even though Category 1, covering transport and sustainable development, has a larger number of posts, they do not have the same impact as the posts belonging to Category 2, referring to transport and climate change. Tweets on topics related to climate change in general are more popular compared to tweets related to the specific policy priorities that can be implemented to tackle climate change, such as autonomous and electric vehicles. The number of retweets indicates that the volume of messages published in a particular category does not guarantee a wider audience if the messages are not attractive enough to captivate the audience's attention, and thus be retweeted and reach even more people in the Twitter social network.

### 4.3. User Category Analysis

In the context of the present study, the sample of "tweets" was categorized based on user type, dividing the initial sample into two (2) categories: regular "Users" and "Policy Influencers". "Policy Influencers" refers to Twitter users with verified accounts (indicated by a blue verified badge) [89]. This type of user is considered to represent official organizations or high-profile accounts (with many followers) whose opinions can reach a significant portion of society. Users who do not have a blue verified badge are classified as "Users". The "Policy Influencers" are categorized into three groups:

- Media, which includes official news websites.
- Industry, which consists of industrial organizations and firms.
- Policymakers representing official political bodies and organizations, such as the European Commission, as well as their official representatives, such as members of the European Parliament.

According to sample statistics, 90% of tweets were published by users, 7% by policymakers, 2% by the industry, and 1% by media sites.

As can be concluded from Figure 4, it was found that users, industry representatives, and policymakers tend to publish more content on topics related to transport and climate change. However, media sites tend to publish more about the different transport policy priorities that can be implemented to tackle climate change. Conversely, users, media sites, and policymakers publish less about transport and sustainable development topics, while industry representatives talk about issues related to policy priorities to tackle climate change.

It is interesting to note that policymakers tend to receive the highest average number of retweets per post, rather than media sites, which one might expect considering their popularity and large audience. Conversely, users tend to receive the lowest number of retweets for their posts. According to the analysis, industry representatives and media sites have a higher average number of followers compared to users. This suggests that these user categories are more popular among the audience.

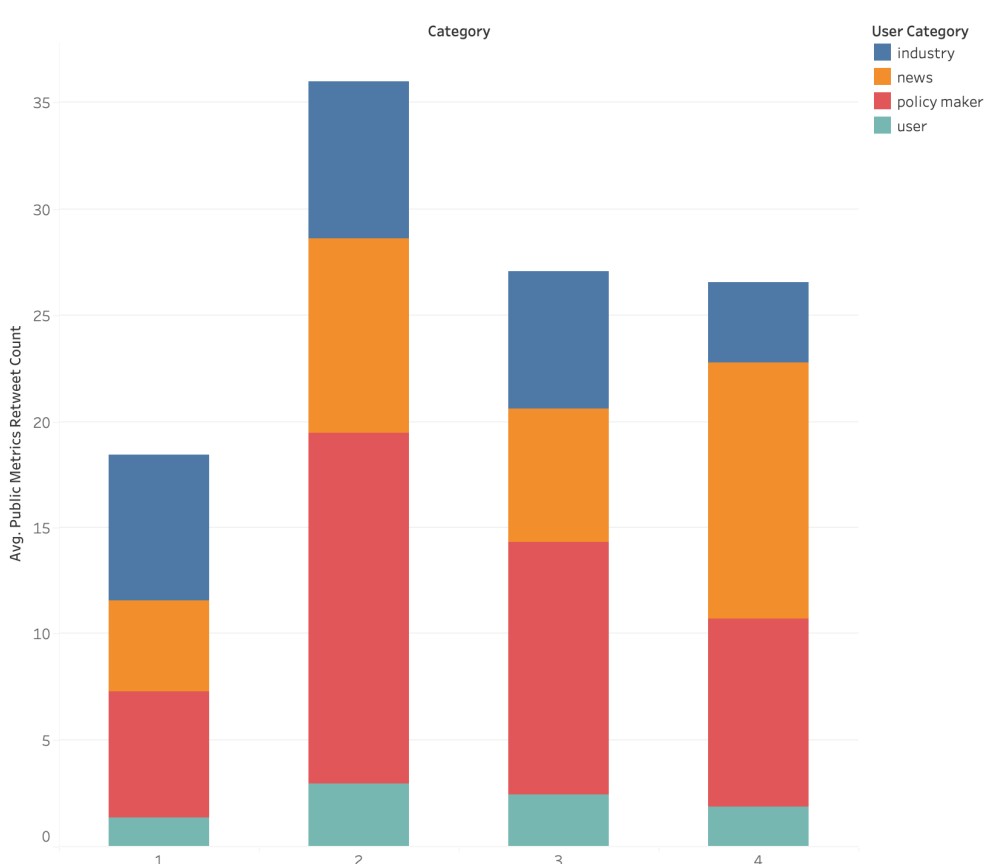

**Figure 4.** Average number of retweets per data and user category.

### 4.4. Topic Modeling

4.4.1. Topic Analysis

After the determination of the optimal hyperparameters for the UMAP and HDBSCAN models, the BERTopic model was initiated and trained on the collected tweet documents. Regarding the outlier reallocation strategy, we decided to utilize the embeddings strategy, as it provides a more suitable representation of the context of each document. Other methods were also tested, with the best alternative being the c-tf-idf method [90], which assesses the frequency and the uniqueness of the terms in a particular document and tests these features against the terms of the identified topics. Although this method provided significantly faster computation time compared to the embeddings method, results were not as accurate, since the sole consideration of term frequency does not provide information on the meaning/context and can thus provide misleading results.

Our topic model ultimately identified 10 unique topics in our database. Table 2 provides an overview of the topics with details on the number of tweets per topic, as well as a title that reflects the thematic field discussed by the tweets in each topic. The topic title is a descriptive narrative that connects the identified terms with specific policy priorities identified in the analyzed EU policy documents, as mentioned in Section 3.2, and other EU or national initiatives. For example, the terms identified in Topic 1 were linked to the New EU Urban Mobility Framework's priority: "Towards climate-neutral cities: resilient, environmentally friendly and energy-efficient urban transport" and the narrative: "Given the availability of suitable technological and other policy solutions for zero-emission mobility, cities should take measures to facilitate the green transition and ultimately ensure that urban mobility and transport becomes climate neutral as early as possible." [76], as well as the European Climate Pact's reference to "Green Mobility" [91]. Another example, which refers to a national initiative, is Topic 6 including terms directly related to UK's strategy for decarbonizing the transport sector [92].

**Table 2.** Number of tweets and description of identified topics.

| Topic | Number of Tweets | Topic Title |
|---|---|---|
| 1 | 15,113 | Sustainable mobility and low-carbon transport for green cities |
| 2 | 14,001 | Innovative mobility solutions for smart and sustainable cities |
| 3 | 8616 | Clean fuel, green hydrogen, and zero-carbon energy transition |
| 4 | 7978 | Shift to public transport to reduce emissions, global warming, and climate change |
| 5 | 3139 | Electric vehicles and charging infrastructure for low-carbon transport and sustainability |
| 6 | 3081 | United Kingdom (UK) Transport Decarbonization Plan |
| 7 | 1905 | Promoting micromobility and active travel choices for low-carbon transport |
| 8 | 1535 | Alternative fuels for decarbonizing transport |
| 9 | 1344 | Decarbonization of the European logistics |
| 10 | 864 | Clean energy and sustainable mobility for climate neutrality in EU |

Figure 5 presents the 15 most representative terms for each of the identified topics, thus giving an abstract description of the context of the focus of each topic. According to the identified terms, Topic 1 focuses on the sustainability and "greening" of transport systems. Additionally, the presence of the terms "cities" and "urban" indicates a focus on urban areas. Topic 2 refers to another strategic priority of the EU policy for urban mobility, i.e., new mobility solutions and transport innovation for smart cities. According to the summary provided in Figure 6, one of the most prominent topics is Topic 3. An overview of the most salient terms of the topic reveals a clear reference [28] to the adoption of hydrogen as an alternative energy source to fossil fuels and the contribution towards limiting emissions from transport. In Topic 4, the impact of the transport sector on climate change and global warming is discussed. More to the point, the presence of terms such as "public transport", "air", "climate", "global warming", and "cars" suggests the "need" to take "action" for accelerating the shift from private to public transport and encouraging the application of policies to this agenda. Topic 5 clearly highlights the role of electrification towards low-carbon mobility. Topic 6 explicitly refers to the UK Transport Decarbonization Plan, which was published towards the end of the reference period of the current research [92,93]. Topic 7 covers another strategic priority of the EU policy for sustainable urban mobility and climate-neutral cities, i.e., the promotion of walking, cycling, and micromobility. The need for clean fuels and renewable energy is discussed in the context of Topic 8. Topic 9 refers to freight transport and logistics, with a focus on the maritime and rail sectors. Finally, a broader reference to the energy transition of the transport sector and its contribution to the climate neutrality of EU is made in the framework of Topic 10. All identified topics are related to the priorities of the current EU transport policy framework and the corresponding objectives of the EU strategic framework, i.e., the Green Deal, as discussed in Section 3.1. The only exception is Topic 6, which is, however, linked to a national policy that is related to the decarbonization agenda.

4.4.2. Topic Evolution and Spread Patterns

The analysis of each topic's semantics provided significant insight into identifying the context of the analyzed tweets and their association with the EU policies. However, the determination of how each topic evolves over time and the assessment of its impact on social media users could provide additional information to practitioners on how relevant topics evolve and disseminate.

As can be derived from the results of Figure 7, there is a gradual increase over time in the number of tweets regarding all of the identified topics. Although Topic 1 was not the most prominent topic for the period between 2011 and 2020, there appears to be a significant increasing trend in the number of tweets associated with it since 2017, which made it the most discussed topic in the years 2020 and 2021. On the other hand, although Topic 2 was the most discussed topic throughout the analysis period, reflecting the popularity of smart mobility applications, results indicate a slight reduction in the number of tweets regarding this topic since 2019. Regarding Topics 3 and 4, although their evolution pattern

is rather similar, with both topics increasing their popularity after 2018, Topic 3 shows a significant increase since 2020, something that reflects the response of the social media community to the adoption of the EU's strategy on hydrogen [91]. Finally, Topic 5 shows a constant increase in popularity over the years, with an increasing trend in 2020 and 2021, something that coincides with the publication of the UK's strategy on the decarbonization of the transport sector [92,93]. The rest of the identified topics present a similar trend over the years, with an overall increase in popularity since 2017.

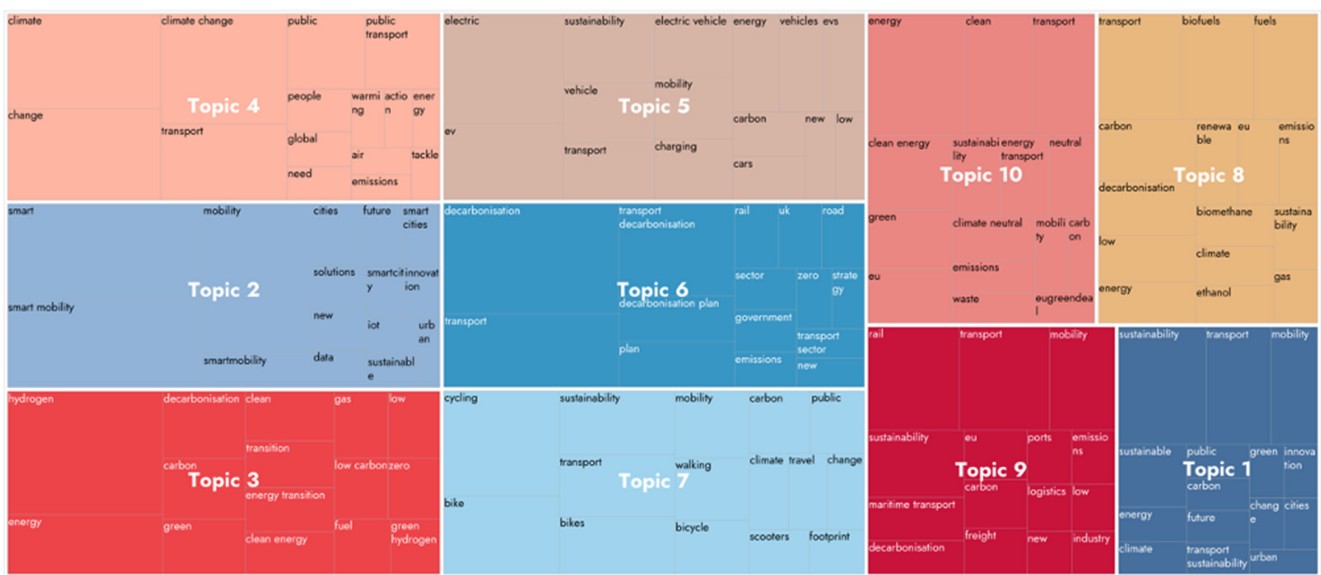

**Figure 5.** The most salient terms for each of the identified topics.

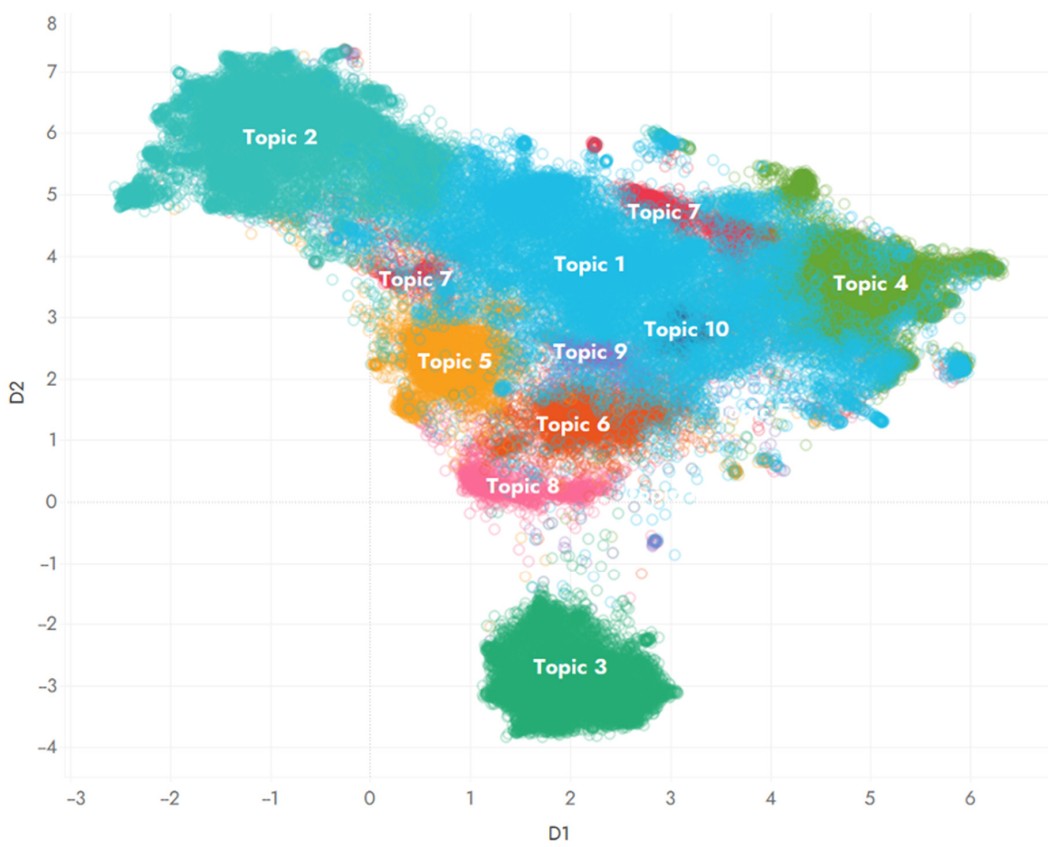

**Figure 6.** Distribution of identified topics based on generated embeddings.

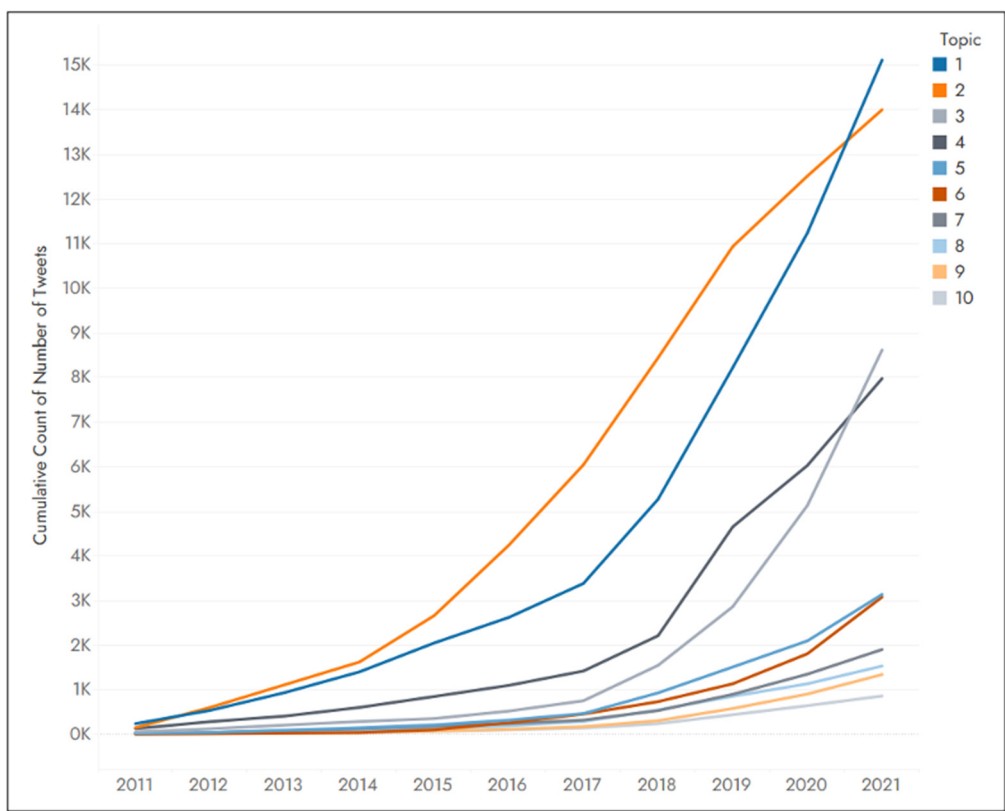

**Figure 7.** Temporal evolution of identified topics for the analysis period.

In order to determine the impact of the subject of each identified topic on the wider social media community, we assessed the average number of retweets for each topic. According to the results of Figure 8, although Topics 1 and 2 are the most prominent in terms of sheer tweet count, they remain relatively low when it comes to being shared by social media users. On the other hand, the energy transition of the transport system towards climate neutrality in EU (Topic 10) appears to be the most impactful subject, with an average retweet number of 6.6 times per tweet. Moreover, the role of public transport and private vehicles in climate change (Topic 4) is also among the most prominent in terms of impact, followed by topics focusing on the promotion of sustainable transport modes (Topic 7) and on UK's strategy for decarbonizing the transport sector (Topic 6).

Although the number of retweets is a representative metric in the assessment of the impact of social media content, in many cases user category is also a significant parameter, as key players in a social media network might influence users more effectively. Figure 9 presents a disaggregated view of the number of retweets per original tweet as well as the most influential users. Results indicate policymakers (political figures, political organizations, and EU organizations) as the most influential user category as they are associated with a higher retweet count across all identified topics. Additionally, simple users (journalists, political party leaders, and environment/mobility advocates) are also associated with impactful content, as the results for Topics 1, 4, and 7 suggest.

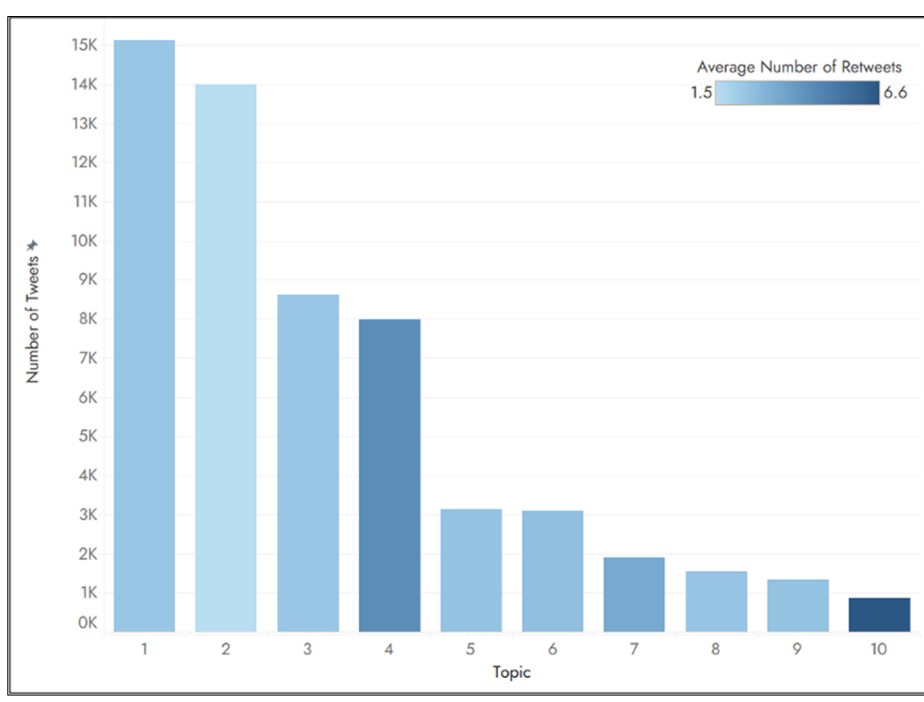

**Figure 8.** Number of tweets and average number of retweets per topic.

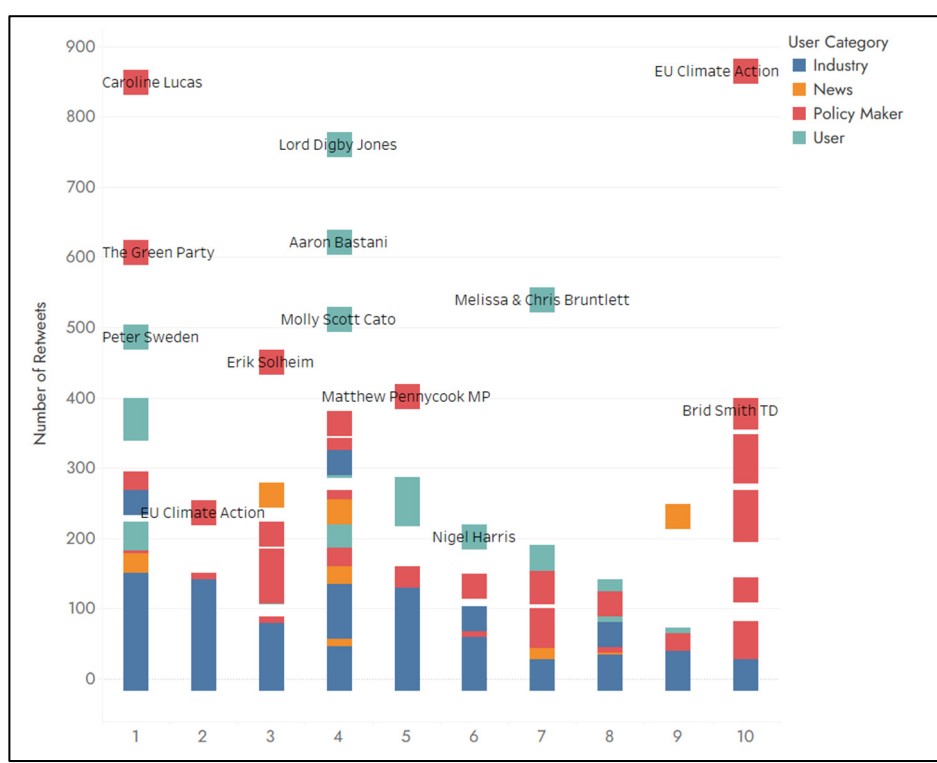

**Figure 9.** Number of retweets per topic and user category.

*4.5. Social Media Influence Analysis*

In this section, we present the outcomes of the network analysis conducted for each data category. We calculated several measures to determine the users' popularity and their impact on spreading a message. We used the NodeXL Pro tool for both network analysis and visualization, as well as the computation of the relevant popularity metrics.

The results of the network analysis are presented in the form of social media network diagrams for each data category (Figure 10). Each network node (Twitter user) is assigned

a color based on the user category. The users are grey, while the policy influencers are blue. Each node's size represents the user's influence, measured by eigenvector centrality. Larger nodes represent higher eigenvector centrality, indicating many connections to other well-connected people. Betweenness centrality is proportional to node opacity, which emphasizes users with high betweenness centrality acting as connectors in a social network for the dissemination of information even to remote network users. The top 10 individuals with the highest number of followers are labeled for identification purposes.

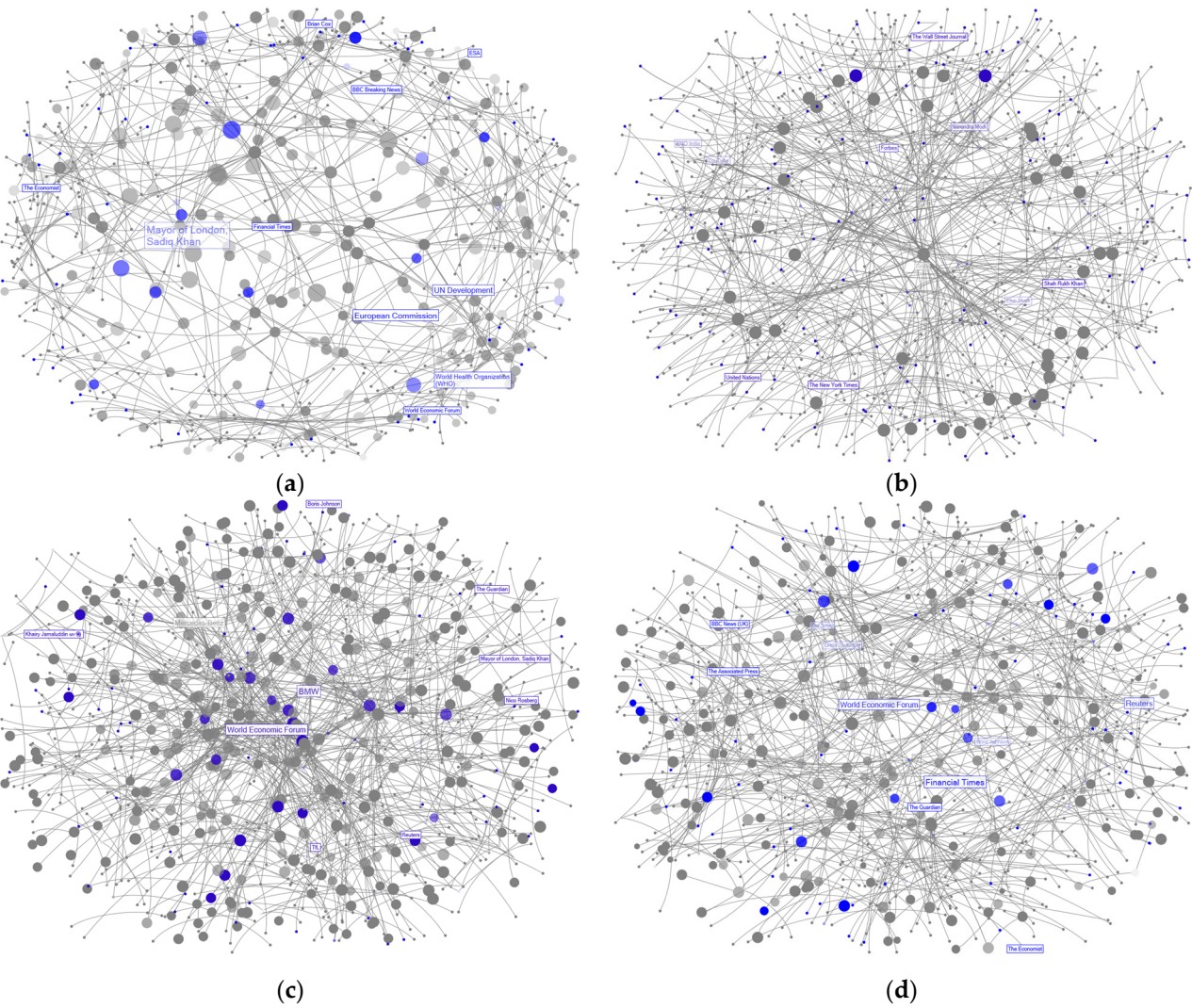

**Figure 10.** Social network analysis diagrams: (**a**) Category 1; (**b**) Category 2; (**c**) Category 3; (**d**) Category 4.

The analysis of the above diagrams reveals that users and policy influencers play a crucial role in transmitting information, regardless of the data category. While one might expect policy influencers and representatives of organizations/companies having an international reputation to have significantly higher popularity than users, this is only sometimes the case. Although policy influencers have a higher number of followers than users (as seen from the labels of the top 10 individuals with the highest number of followers), the size and opacity of the nodes suggest that the number of followers alone is not enough for a user to effectively transmit information on a social network like Twitter. On the other hand, users have strategic positions in conveying information in the network, as evidenced by their eigenvector centrality and betweenness centrality. Eigenvector and betweenness centrality metrics are two methods used to measure how engaged a user is with other users

and their content on social media platforms. They provide insights into the actual attention given to content and the actions users take to disseminate information. This is different from the number of followers a user has on Twitter, which only indicates the potential reach of a tweet. A high number of followers does not necessarily mean all followers read tweets. However, it does indicate a certain level of popularity. A critical distinction between follower and centrality metrics is that the former measures centrality within the entire Twitter network, while the latter measures centrality within the topic-specific network. According to the above, while policy influencers hold a significant position in the Twitter network, users are crucial in spreading information and promoting EU climate and energy policy priorities related to transport and mobility.

Upon analyzing the different data categories, it was noticed that in all categories except Category 2, users and policy influencers are crucial in transmitting information. However, in Category 2, very few policy influencers contribute to disseminating information related to transport and climate change, compared to users who play a decisive role. Regarding the most significant policy influencers, both in terms of popularity (number of followers) and central position in the network, Category 1, involving transport and sustainable development, includes mainly organizations such as the European Commission, the United Nations, and the World Health Organization, along with journalistic sites such as the BBC and the Financial Times. In Category 2, only news agencies are listed apart from the United Nations. In Category 3 (transport and policy priorities for tackling climate change), besides international organizations such as the World Economic Forum and news agencies such as The Guardian, the BMW company plays a vital role in this network, demonstrating the interest of the automobile industry in issues related to this category. In addition, in this category there are a significant number of nodes, both users and policy influencers, with characteristics that can favor the wide transmission of information of topics related to the specific category. Lastly, Category 4, referring to specific climate and transport policy priorities, is dominated by policy influencers from the news sector.

## 5. Discussion

For over three decades, EU transport policy has been geared towards promoting environmental sustainability and energy efficiency. However, since 2015, European policymakers have progressively honed in on specific sustainable development goals: decarbonizing the transport system and transitioning to zero-carbon technologies and renewable fuels. This strategic shift aligns with the EU commitment towards ambitious global climate change targets. Furthermore, Europe recognizes the pressing need to bolster energy efficiency and resilience against potential disruptions. Leveraging unprecedented technological innovation is seen as instrumental in achieving these objectives, offering transformative opportunities to enhance energy efficiency and mitigate the carbon footprint of the transport sector.

To effectively communicate policy priorities and gather feedback from stakeholders, EU policymakers are increasingly utilizing social media, notably the Twitter platform. This fosters dynamic discussions among policymakers, industry stakeholders, the media, and the general public regarding the role of transport policy in addressing climate change and energy transition challenges.

In this context, the current research undertakes data mining and topic modeling techniques to track the evolution of user-generated content on Twitter, which has been instrumental in fueling these discussions across Europe from 2011 to 2021. Moreover, both the content and the platform users were categorized to analyze and identify emerging topics throughout the dataset. Additionally, the analysis aims to understand how discussions propagate through the social network and identify key influencers who play pivotal roles in disseminating information within topic-specific networks. The current research is based on open data and implements robust text-mining and analysis techniques in combination with a comprehensive EU policy review. In this way, a low-cost methodological approach is proposed to assess the effectiveness of EU policy dissemination though social media,

which can be easily adapted to similar research purposes regardless of the examined policy domain.

Regarding the first research question of the present study, the analysis clearly indicates a substantial increase over time in the volume of tweets pertaining to the relationship between transport, mobility, climate crisis, and energy transition. This trend is attributed to both the growing usage of the specific social media platform and the escalating significance of these aspects within the broader transport policy agenda. Furthermore, the discussion evolves from a general exploration of the interaction between transport and sustainability during the first half of the analysis period to more specific priorities concerning vehicle, engine, and fuel technologies aimed at addressing decarbonization and energy transition targets in the second half. This evolution mirrors the heightened global interest in climate-related policies and the rapid advancements in relevant technologies. Regarding the thematic areas covered by users' content based on the second research question, the current topic analysis indicated a variety of topics, with the most predominant ones focusing on the sustainability of the transport sector, the adoption of hydrogen as an alternative fuel source, and the utilization of smart mobility applications towards decarbonization and energy transition. Furthermore, a significant part of the analyzed tweets concentrates on the importance of increasing public transport use as a measure to mitigate the effects of climate change, while electrification and its role in the EU's decarbonization strategy is also highlighted.

The analysis by user category uncovered distinct trends in content publication. Users, industry representatives, and policymakers predominantly focus on topics related to transport and sustainable development in general. Conversely, media sites tend to publish more content related to climate change, global warming, and carbon footprint issues. These findings underscore differing priorities among user categories, with stakeholders in transport and sustainable development emphasizing broader themes, while media outlets prioritize environmental concerns and climate-related topics.

The findings suggest that policymakers, including political figures, political organizations, and EU institutions, emerge as the most influential user category, as evidenced by their higher retweet counts across all identified topics. Furthermore, simple users, such as journalists, political party leaders, and advocates for environmental and mobility issues, also wield significant influence, particularly in topics related to sustainable mobility, global warming, and climate change.

To answer the third and final research question, social network analysis was performed on the data collected. The social network analysis highlights the crucial roles played by both users and policy influencers in disseminating information on platforms like Twitter. While policy influencers may boast a higher number of followers, this metric alone does not ensure effective information transmission. Metrics such as eigenvector and betweenness centrality offer insights into user engagement and the attention garnered by content. Users strategically positioned within the network play pivotal roles in conveying information and are instrumental in promoting EU climate and energy policy priorities related to transport and mobility. Identifying the most influential users across the network is essential. These users possess the capability to transmit information to a vast audience, as their posts are not only visible to those with whom they directly interact but also to their followers. The potential reach of information expands exponentially as the number of users interacting directly with a key influencer increases. Hence, these core users are ideal candidates for spearheading campaigns to promote European policies and initiatives via Twitter, ensuring that the intended message reaches the widest possible audience.

The findings represent significant evidence regarding the progression of EU transport policy communication in addressing climate change and energy transition, closely intertwined with pivotal policy documents and milestones. EU policymakers gain a comprehensive understanding of how proposed policies are disseminated across various sectors, including industry and media, and embraced by the wider populace. Ultimately, this research lays a foundational framework for developing monitoring and evaluation mecha-

nisms at both EU and national levels. These mechanisms are imperative for systematically assessing the efficacy of communication strategies in conveying strategic priorities across sectors to effectively mitigate and adapt to climate change.

Our research also holds practical implications of significance. For instance, it underscores the importance of considering social media as a pertinent social context for gauging trends related to climate change and the challenges of energy transition. This understanding is crucial for devising communication campaigns aimed at promoting the adoption of low-emission mobility solutions and renewable or alternative fuels to decarbonize the transportation system.

Furthermore, our data highlight the necessity for EU organizations to carefully consider the diverse roles that various types of users can play in fostering awareness and engagement regarding EU climate and energy policy priorities for transport and mobility. While previous studies have noted the significant contribution that traditional media can make in enhancing public understanding of sustainability issues and raising awareness about climate change, it appears that awareness-raising campaigns can be more effective when employing multiple communication strategies. Combining social media platforms with traditional mass-media channels has been identified as one of the most promising approaches in this regard.

Therefore, designers of public education campaigns should recognize social media's dual role as both a source of information on evolving societal trends and a social arena capable of facilitating community involvement in pro-environmental behaviors. With effective management, social media networks can serve as powerful tools for building relationships, fostering engagement, and promoting involvement in specific topics. Consequently, we recommend that governmental institutions worldwide increase their interactions through these channels as well.

Social media discussions have proven instrumental in driving policy changes, particularly in the mobility sector. Social media plays a crucial role in influencing policy decisions by raising awareness, shaping public opinion, and mobilizing advocacy efforts. Social media creates public pressure through various mechanisms and compels policymakers to respond to societal needs. It provides a feedback loop for policymakers to gauge public opinion and adapt policies accordingly. Social media also fosters political accountability by enabling citizens to scrutinize policies and drive transparency. Social media discussions have significantly impacted policy changes in the mobility sector. For example, social media has shaped regulations for ride-sharing services such as Uber and Lyft, as policymakers have been prompted to address safety and labor rights concerns raised on these platforms [94]. In addition, social media campaigns have brought about infrastructure planning decisions, including allocating resources for bicycle lanes and pedestrian-friendly infrastructure [95]. Furthermore, advocacy efforts on platforms like Reddit and Twitter have promoted electric vehicle adoption and helped expand charging infrastructure [96]. Social media discussions have also influenced micromobility regulations, leading to safety standards and designated parking zones for electric scooters and shared bicycles [97]. The above examples illustrate how discussions on social media platforms can play a crucial role in increasing people's knowledge, shaping their opinions, mobilizing advocacy efforts, and providing policymakers with feedback. This, in turn, can drive policy changes with regard to transport policy's role in addressing climate change and energy transition challenges.

This study also has several notable limitations that warrant attention in future research endeavors. Firstly, focusing solely on one social media platform may have resulted in a narrow perspective of the social phenomenon under investigation. While Twitter is one of the most widely used social media platforms, it does not encompass all online discussions, and individuals may prefer alternative platforms. Consequently, the data collected may not fully represent all social media conversations during the study period, potentially limiting the generalizability of our findings to the specific characteristics of Twitter.

Moreover, the selection of a specific social media platform could have impacted the diversity of topics discussed and the viewpoints expressed. For instance, Twitter is

typically recognized for its focus on broader societal issues and public discourse, in contrast to platforms such as Facebook and TikTok, which are often utilized for more personal or intimate interactions. This inherent bias may have resulted in an overemphasis on topics related to environmental and societal concerns, neglecting more individual perspectives. To mitigate this limitation, future investigations could broaden their scope by incorporating a wider array of social media platforms, including those commonly used for sharing personal thoughts and experiences.

Furthermore, future research endeavors could delve into alternative communication modalities beyond textual comments. Emoticons, photos, and videos have the potential to convey nuanced perceptions and emotions just as effectively as textual content [98]. Therefore, adopting novel approaches to analyze these diverse communication modalities would enrich our understanding of social media discourse and provide a more comprehensive insight into public sentiments and attitudes.

## 6. Conclusions

This research endeavor is poised to contribute significantly to our understanding of how European public organizations leverage social media platforms, particularly Twitter, to engage with citizens and communicate policy-related information. By focusing on themes such as climate change, energy transition, and smart mobility, this study addresses critical issues facing society today, while also shedding light on the evolving landscape of digital communication in the public sector.

Our findings underscored the pivotal role of social media platforms, particularly Twitter, in facilitating dynamic exchanges among policymakers, industry stakeholders, media outlets, and the public on key policy priorities. Notably, the discourse evolved from general explorations of sustainability to more focused discussions on decarbonization strategies and renewable energy solutions, mirroring global interest and technological advancements in these areas. Crucially, we identified distinct trends in content publication across different user categories, with policymakers emerging as the most influential group. However, journalists, political party leaders, and advocates also wielded significant influence, particularly in topics related to sustainable mobility and climate change. Social network analysis revealed the critical roles played by both users and policy influencers in disseminating information and shaping discussions on Twitter. Identifying key influencers is vital for amplifying EU policy priorities and initiatives to a broader audience, thus enhancing engagement and awareness.

The integration of text-mining techniques, including topic modeling, and social network analysis, represents a methodological strength that allows for a comprehensive exploration of social media data. Topic modeling facilitates the identification of key themes and discussions within social media conversations, enabling researchers to uncover emerging issues and trends. Additionally, social network analysis offers valuable insights into the structure of online communities, the dissemination of information, and the influence of key actors within social media networks.

By adopting a multidimensional approach that combines quantitative analysis with qualitative insights, this study aims to develop a nuanced understanding of how European agencies engage with citizens on social media platforms. The proposed investigation into social media metrics and performance measurement further enriches the research by addressing the need for robust evaluation frameworks tailored to the unique goals and objectives of public organizations.

Despite the valuable insights gained, this study has limitations, including its exclusive focus on Twitter and the potential bias inherent in platform selection. Future research should expand to include a broader range of social media platforms and explore alternative communication modalities such as emoticons, photos, and videos to provide a more comprehensive understanding of social media discourse.

Ultimately, this research endeavor has the potential to inform strategic decision-making within European agencies, offering actionable recommendations for optimizing

social media strategies, enhancing citizen engagement, and improving communication effectiveness. By bridging the gap between theory and practice, this study contributes to the broader discourse on digital governance, public communication, and policy implementation in the context of contemporary societal challenges.

**Author Contributions:** Conceptualization, A.N., A.K., N.G. and I.P.; methodology, A.N., A.K. and N.G.; software, A.K. and A.N.; validation, A.N., A.K. and N.G.; formal analysis, A.K. and A.N.; investigation, A.K., A.N. and N.G.; data curation, A.N. and A.K.; writing—original draft preparation, A.N., A.K., N.G. and I.P.; visualization, A.K. and A.N.; supervision, I.P.; project administration, A.N., A.K. and I.P. All authors have read and agreed to the published version of the manuscript.

**Funding:** This research received no external funding.

**Institutional Review Board Statement:** Not applicable.

**Informed Consent Statement:** Not applicable.

**Data Availability Statement:** The data presented in this study are available on request from the corresponding author.

**Conflicts of Interest:** The authors declare no conflicts of interest.

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
