# Peer review of "Assessing the EU Climate and Energy Policy Priorities for Transport and Mobility through the Analysis of User-Generated Social Media Content Based on Text-Mining Techniques"

_sustainability, doi:10.3390/su16103932_

Round 1
Reviewer 1 Report
Comments and Suggestions for Authors
In order to improve the quality of this study, some comments have been given as below.
1.Please add a subsection of literature review. The current version cannot provide sufficient information for readers to know the development of related works about using text mining in social media.
2.In the step of pre-process data, it's not clear. Authors should mention (1)how to crawl the text data (2)how to do natural language processing (NLP), including tokenization, removing stop words, lemmatization, and so on in detail.
3.In table 1, how did authors find keyword and how the categorize the collected data should be clarified.
4.In section 2, authors should clarify the used methods and implemental procedure step by step according to Figure 1. The current version is difficult for readers to follow.
5.In subsetion 2.4, authors used BERT. Authors should clarify parameters settings and explain how use BERT to extract topics.
6.Section 2 should introduce the procedure of used methods. But, subsection 2.5 merely introduce used tools. Authors are suggest to clarify the step of doing social networking analysis, instead of introducing used software.
7.How did authors determine the optimal number of selected topics?
8.Authors should explain how to name these selected topics.
Comments on the Quality of English LanguageEnglish needs minor revision.
Author Response
Dear reviewer,
Thank you for your comments. Please find our replies in the attached document.

Reviewer 2 Report
Comments and Suggestions for Authors
After reviewing the paper "Assessing the EU climate and energy policy priorities for transport and mobility through the analysis of user-generated social media content based on text-mining techniques," it's clear that this study provides a detailed and relevant exploration of the ongoing discussions around EU transport policies concerning climate change and energy transition on social media platforms like Twitter. By employing robust methodologies that combine text mining and social network analysis, the paper delivers a thorough insight into public engagement and the dissemination of policy-related discussions over the last decade. The innovative use of analytical tools such as BERTopic for topic modeling and NodeXL for social network analysis significantly contributes to the fields of policy communication analysis both academically and practically.
The paper shines in its empirical approach, offering actionable insights into how discussions on social media mirror and potentially impact policy communication strategies. Yet, to augment its depth, impact, and clarity, the paper could benefit from a few enhancements:
Firstly, integrating a more robust theoretical framework to better connect social media analysis with policy decision-making and public engagement theories could enrich the study's context. Such a framework could include models like the Sphere Model of Public Discourse to delineate how social media influences policy discussions, or delve into the policy-making process with theories like multi-flow models or policy cycle models, to examine how social media discussions impact different stages of policy-making.
Furthermore, establishing a clearer causal relationship between social media discussions and policy changes presents a challenge. The complexity of the policy-making process, influenced by various factors and stakeholders, makes it difficult to precisely measure the impact of public discourse on social media on actual policy decisions. Strengthening the connection between social media sentiment and policy outcomes would offer deeper insights into the influence of public opinion on policy formulation, enhancing the understanding of the interplay between online discourse and policy changes.
Given these considerations, the recommendation is for 'Revision and Resubmission.' This decision reflects the potential of the paper to make a substantial contribution to the field, provided that the suggested enhancements are carefully addressed. The revisions should aim to provide a more comprehensive theoretical grounding and a detailed examination of the linkage between social media sentiment and policy outcomes. I look forward to seeing the revised manuscript and believe that these improvements will make a valuable addition to the academic discourse on policy communication and public engagement.
Author Response

(The authors gave the same response as above.)

Reviewer 3 Report
Comments and Suggestions for Authors
Thank you for the opportunity to review an interesting article analysing how EU transport and mobility policies with climate change and energy transition issues are communicated to stakeholders through Platform X.
Before publication, the article requires needs minor improvement.
The manuscript is written in the correct scientific language. The Authors precisely formulated the purpose of the research as well as clearly described the research methodology and results. However, I have made a few comments that authors should take into account before publishing the manuscript
Lines 57 and 116: The text „Click or tap here to enter text” should be deleted.
A direct reference to the purpose and the three research questions should be given in the 'Discussion' section.
The chapter 'Conclusions' should be rewritten. Some of the content in this chapter corresponds more to the 'Discussion' section than to the conclusions. The conclusions of the research should be presented in a precise, brief way, as bullet points.
Author Response

(The authors gave the same response as above.)

Reviewer 4 Report
Comments and Suggestions for Authors
This manuscript is aimed at exploration of changes in user-generated content in "X" network, connected with EU climate and energy policy. The period of analysis 2011-2021. From my point of view, such studies are interesting and important to understand the mood of public. At the same time i have some concerns, which should be considered before the final decision.
Critical comments:
1.I don't see significant contribution of this manuscript to research field. Almost all methods are in-box solutions of python libraries. Please, explain and clarify the scientific novelty.
2.Conclusion should be connected with the description of specific practical recommendations. Right now i see only the analysis without clear output.
Minor comments:
1.It is necessary to use official title of social network - X. You can add that it was formerly known as Twitter, but it is not correct to use previous title. On the other hand, you can add explanation that you use previous title, because you use data, generated before the rename. Please, choose one of these ways.
2.All acronyms (also in figures) should be deciphered at the first mention.
3.Lines 57 and 116. "Click or tap here to enter text.." What does it mean?
4.Lines 120-129. It would be better to write this paragraph in a past tense, because now i see news like https://deadline.com/2023/05/twitter-faces-potential-ban-europe-elon-musk-company-pulled-back-disinformation-rules-1235382412/
5.Line 205. "two (2)". Please, confirm that it is not a repeat.
6.Lines 234-248. If all values were calculated, is it possible to provide equations for all calculations?
7.Figures:
7.1.Fig. 5. It is almost impossible to read it.
7.2.Fig. 7. Legend is necessary.
7.3.Fig. 8, 9. Please, add the title of horizontal axis.
7.4.Fig 10. What about c and d?
Comments on the Quality of English Languageseems fine, but minor polishing is required
Author Response

(The authors gave the same response as above.)

Round 2
Reviewer 1 Report
Comments and Suggestions for Authors
All of my comments have been fully considered in this version. I think it could be accepted.
Author Response
Dear reviewer,
Thank you for your valuable comments throughout the reviewing process, which have enhanced the quality of our paper.

Reviewer 2 Report
Comments and Suggestions for Authors
The revisions to the manuscript have notably addressed my initial feedback by incorporating a new section in the literature review that examines the use of social media by governmental entities and the consequent implications. Furthermore, the discussion around text mining techniques, and their integration with policy decision-making and public engagement theories, aligns well with the enhancements I recommended. However, while the paper now better connects the theoretical aspects of social media influence on policy-making, it still does not provide a clear demonstration of how social media discussions translate into actual policy changes. This aspect was crucial in my original comments, highlighting the need for a more explicit elucidation of these causal relationships. To address this gap effectively, I recommend incorporating more detailed examples, or case studies that could substantively illustrate these dynamics. This additional evidence would greatly strengthen the paper's assertions about the impact of social media on policy-making processes.
Author Response
Dear reviewer,
Thank you for your valuable comments which we believe that enhance the quality of our paper. We have addressed your recent comments, as described in the attached Reply to Reviewers_Revision 2. We hope that the revised version of our paper adequately covers the issue that you have raised in your review.

Reviewer 4 Report
Comments and Suggestions for Authors
Authors have made all required corrections.
Author Response

(The authors gave the same response as above.)
